# Low Permeability Gas-Bearing Sandstone Reservoirs Characterization from Geophysical Well Logging Data: A Case Study of Pinghu Formation in KQT Region, East China Sea

**Feiming Gao** [1,2]**, Liang Xiao** [1,2,*]**, Wei Zhang** [3]**, Weiping Cui** [3]**, Zhiqiang Zhang** [3] **and Erheng Yang** [1,2]

1   School of Geophysics and Information Technology, China University of Geosciences, Beijing 100083, China
2   State Key Laboratory of Geological Processes and Mineral Resources, China University of Geosciences, Beijing 100083, China
3   Well-Tech Department of China Oilfield Services Ltd. (COSL), Beijing 100005, China
*   Correspondence: xiaoliang@cugb.edu.cn

**Abstract:** The Pinghu Formation is a low permeability sandstone reservoir in the KQT Region, East China Sea. Its porosity ranges from 3.6 to 18.0%, and permeability is distributed from 0.5 to 251.19 mD. The relationship between porosity and permeability was poor due to strong heterogeneity. This led to the difficulty of quantitatively evaluating effective reservoirs and identifying pore fluids by using common methods. In this study, to effectively evaluate low permeability sandstones in the Pinghu Formation of KQT Region, pore structure was first characterized from nuclear magnetic resonance (NMR) logging based on piecewise function calibration (PFC) method. Effective formation classification criteria were established to indicate the "sweet spot". Afterwards, several effective methods were proposed to calculate formation of petrophysical parameters, e.g., porosity, permeability, water saturation ($S_w$), irreducible water saturation ($S_{wirr}$). Finally, two techniques, established based on the crossplots of mean value of apparent formation water resistivity ($R_{wam}$) versus variance of apparent formation water resistivity ($R_{wav}$)—$S_w$ versus $S_{wirr}$—were adopted to distinguish hydrocarbon-bearing formations from water saturated layers. Field applications in two different regions illustrated that the established methods and techniques were widely applicable. Computed petrophysical parameters matched well with core-derived results, and pore fluids were obviously identified. These methods were valuable in improving low permeability sandstone reservoirs characterization.

**Keywords:** low permeability; sandstone reservoirs; pore structure; fluid identification; formation classification

## 1. Introduction

With the development of oil and gas exploration techniques, exploration targets have transformed from original high-amplitude structure and simple lithologic reservoirs to low-amplitude, low permeability of tight sandstone reservoirs [1,2]. Tight reservoirs became the main battlefield to ensure oil and gas resources supplements [2]. Low permeability to tight sandstone reservoirs accounted for more than 60% of the newly added oil reserves in recent decades in China [3].

The Pinghu Formation in KQT Region, East China Sea, belonged to typical low permeability sandstone reservoirs. It exhibited characteristically high detritus and clay contents, complicated carbonate cements, and mesopores to small pores. Tubular and sheet-like throats dominated the structure [4,5]. It was difficult to quantitatively identify and evaluate such types of reservoirs with common methods [6]. The traditional rock volume model, which was used to calculate formation parameters from well logging data, was poorly adaptable for this purpose. The reasons can be summarized into four aspects: First, matrix parameters to calculate porosity were difficult to acquire due to complicated lithologic type and composition. Second, the relationship between porosity and permeability was

poor due to complicated pore structure and strong heterogeneity; permeability cannot be calculated from porosity based on current methods because they were established in formations with relatively simple pore structures [7,8]. Third, Archie's equation, which was valuable in conventional reservoirs, cannot express electrical conductivity in low permeability sandstones. Relationships between porosity versus formation factor, water saturation and resistivity index are not a simple power function, and fixed cementation exponent *m* and saturation exponent *n* cannot be acquired. Fourth, pore fluids cannot be easily identified by using single resistivity or porosity due to a slight contribution to the logging response. Physical properties differences between oil-bearing reservoirs and water saturated layers are not obvious.

Porosity calculations are as mainly determined by density, neutron and interval transit time, and many models have been proposed [9–12]. In low permeability gas-bearing reservoirs, the geophysical well logging response is affected by many factors, e.g., lithology, pore size and pore fluids. Thus, current models cannot be directly used in our target formations before formation properties are firstly assessed. Permeability calculation always face great challenges in low permeability reservoirs due to complicated pore structure and strong heterogeneity, leading to a weak relationship between porosity ($\varphi$) and permeability ($K$) (Figure 1) [13,14]. Generally, multiple regression statistical methods or hydraulic flow unit (HFU) approaches are ere used to calculate permeability [15–17]. In addition, models of calculating permeability based on nuclear magnetic resonance (NMR), dipole shear wave, resistivity image logging, fractal theory and neural networks have also been established. Good results have been acquired under certain conditions. The wide applicability of these models was not verified [18–22].

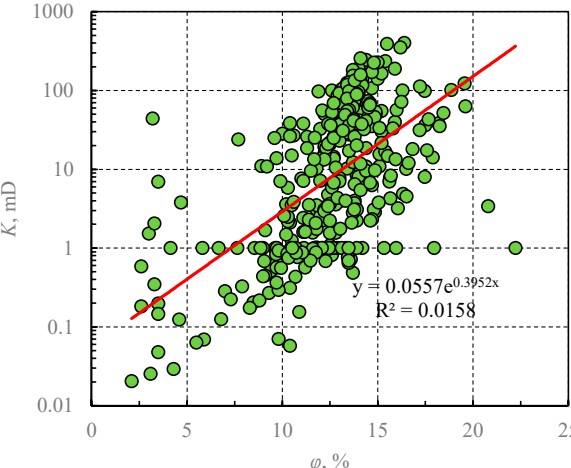

**Figure 1.** Relationship between core-derived porosity ($\varphi$) and permeability ($K$) in the Pinghu Formation of KQT Region, East China Sea. Relationship between these two parameters are poor. This makes it impossible to accurately calculate permeability from porosity.

In conventional reservoirs, water saturation can be calculated from resistivity and porosity well. The calculation is based on Archie's equation and uses the fixed cementation exponent *m* and saturation exponent *n* [23–26]. However, *m* and *n* are divergent in low permeability and tight sandstone reservoirs due to complicated pore structures (Figure 2a,b). To accurately calculate water saturation, many valuable models have been proposed. The Waxman–Smits model and the dual water model, which consider additional conductivity caused by clay minerals, were raised to calculate water saturation in shaly sand reservoirs [27–30]. Givens (1987) and Givens and Schmidt (1988) proposed a conductive rock matrix model (CRMM) to calculate water saturation after introducing additional conductivity of matrix [31,32]. In addition, some empirical equations, such as Simandoux's equation, Indonesia saturation equation and Nigerian saturation formula, were also raised to introduce the effect of shaly to conductivity [33,34]. These models and

equations were all established based on the contribution of shaly or clay to additional conductivity: the impact of pore structure was ignored. In low permeability and tight reservoirs, no specific water saturation calculation equation was available, and Archie's equation was still used [35]. This made greatly decreased the accuracy of the water saturation calculation. Many scholars proposed models to optimize the involved parameters in Archie's equation [36–41]. Mao et al. (2000) and Li et al. (2012) proposed a theoretical model to calculate various cementation exponents from porosity [36,37]. Xiao et al. (2013) used parameters: percentage of macropore and small pore components to characterize low permeability sandstones pore structures. They introduced an optimal saturation exponent calculation model [38]. In addition, Arifianto et al. (2018) pointed out that formations can be classified into several types, and respective values of *m* and *n* should be used to improve water saturation [42]. These methods were established based on resistivity and NMR experiments of core samples; plenty of data should be first acquired. These data are available only in exploration wells where abundant well logging occurs and experimental data can be acquired. However, in development wells, especially in offshore wells, the limitation of data acquisition reduces their applicability.

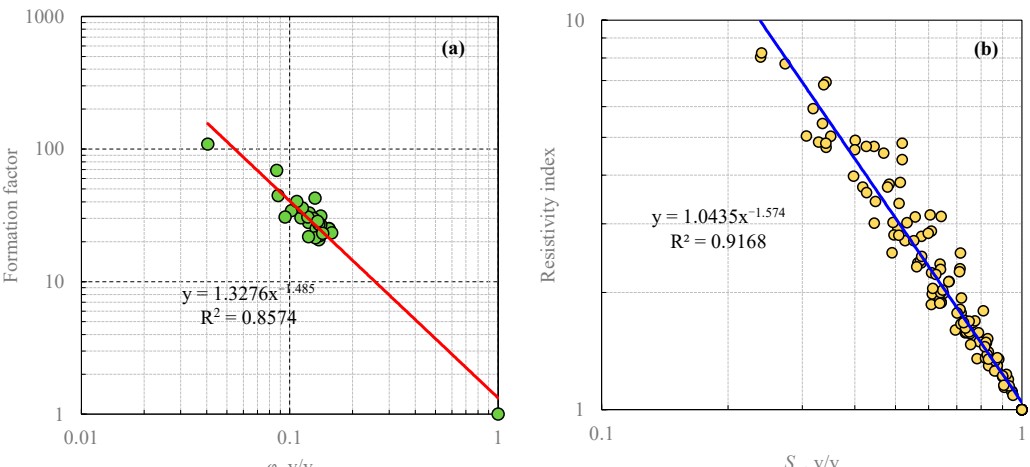

**Figure 2.** Relationship between porosity versus formation factor (**a**) and water saturation versus resistivity index (**b**) in the Pinghu Formation of KQT Region, East China Sea. Divergent relations among these parameters led to low water saturation calculation accuracy based on the traditional Archie equation.

Pore structure characterization is of great importance in improving low permeability to tight sandstone formation evaluation [43]. Generally, the mercury injection capillary pressure (MICP) curve was optimal in characterizing pore structure [44,45]. By using MICP curves, formations can be classified into several types. The best type always corresponds to the highest quality and, thus, good hydrocarbon production [46]. However, formation pore structure cannot be consecutively characterized due to limitation of quantity that is caused by environmental and economic factors [47]. Nuclear magnetic resonance (NMR) logging has unique advantages in consecutively evaluating formation pore structures [48,49]. The common method to evaluate pore structure was to synthetize capillary pressure curves ($P_c$) from NMR logging; several techniques have been raised in the last 12 years [20,43,47,50–52]. Volokitin and Looyestijn (2001) and Looyestijn (2001) proposed a linear scale function to transform the NMR $T_2$ distribution as a pseudo-$P_c$ curve to characterize formation pore structure [43,50]. This function was verified to be available in conventional formation with a high quality. However, in low permeability sandstone reservoirs, pore structure was overestimated [20]. Although Xiao et al. (2016) raised an alternative method to construct pseudo-$P_c$ curve form NMR logging based on formation classification, it had regional limitations [20]. In different regions or for different types of formation, the classification criteria were varied, meaning that it cannot be widely used.

The purpose of this paper was to reach several objectives: (i) characterizing low permeability sandstone reservoirs pore structure from NMR logging based on the improved model; (ii) establishing several effective models to calculate formation petrophysical parameters; (iii) raising the criteria to distinguish low resistivity contrast hydrocarbon-bearing reservoirs from water saturated layers by using geophysical well logging data. Good consistency between calculated results with core-derived results illustrated the reliability of the proposed models. This would be very valuable in improving characterization of our target: low permeability sandstone reservoirs.

## 2. Geological Setting

Xihu Sag is located in northeastern East China, in the Sea Shelf Basin. From west to east, Xihu Sag was divided into three structural sub-units: the western slope belt, the central inversion belt and the eastern fault belt (Figure 3). The Pinghu slope belt is located in the middle of the western slope belt, and it can be further sub-divided into four regions from south to north: the Pinghu, Baoyunting, Wuyunting and KQT Regions. The KQT Region is located in the nose-shaped uplift belt. In the Paleocene–Eocene rifting period, the KQT Region developed contemporaneous faults and structural traps and became the group of fault blocks that had fallen steadily along the trend. The main oil and gas producing layers were Eocene Pinghu Formation [53]. The sedimentary period of the Eocene Pinghu Formation was in a warm and humid climate and in a freshwater environment. The tidal river-controlled delta facies were mainly developed, and two sedimentary models occurred: In the period of low water levels, the sediment was mainly in the river-controlled delta. In the period of transgression, the salinity of the water body increased, and the tides had a certain influence on sediment distribution [54].

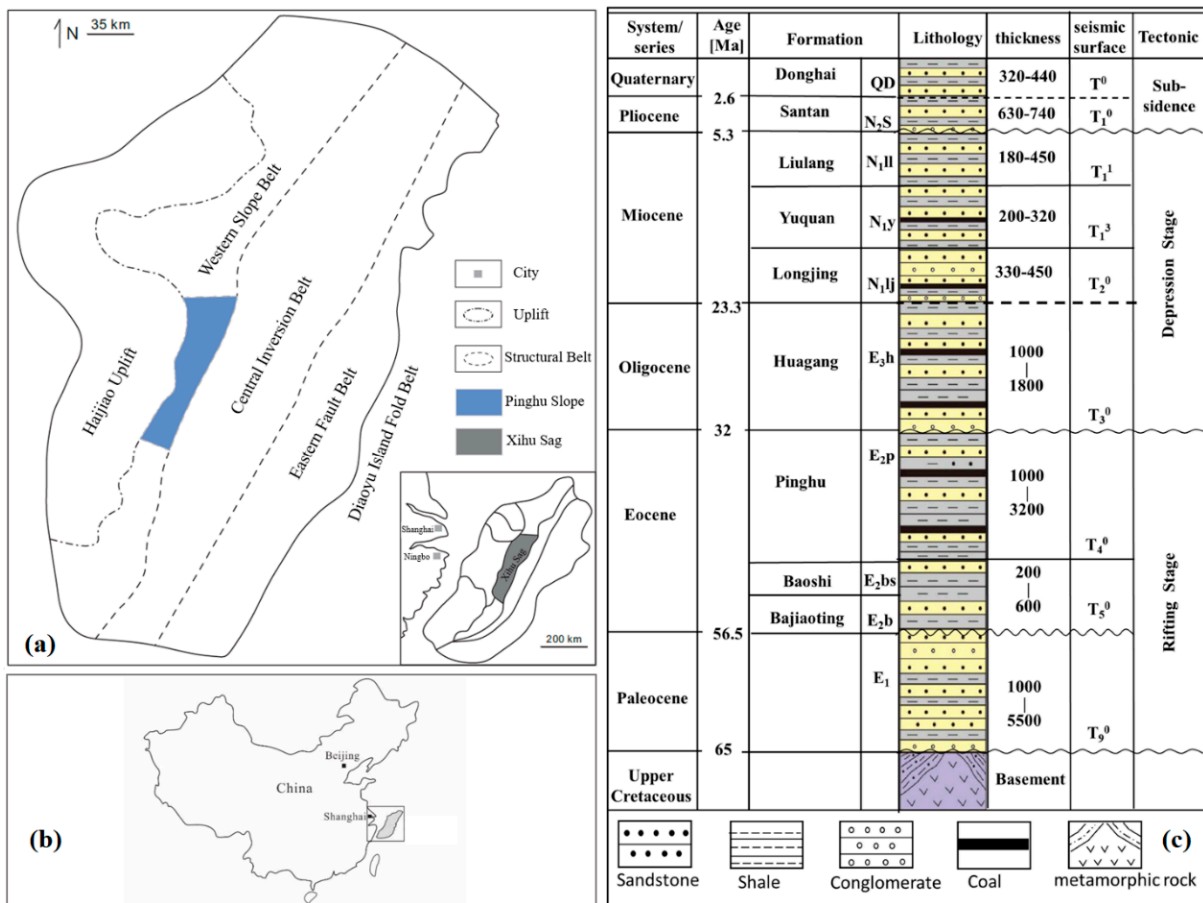

**Figure 3.** Location of the study area in the East China Sea Shelf Basin (**a**), location of the East China Sea Shelf Basin in east China (**b**), and the formation member, thickness and lithology (**c**) [53,55].

## 3. Reservoir Petrophysical Characteristics

Mineral composition, grain cementation and arrangement determine reservoir quality, as well as porosity and permeability [56]. Petrophysical characteristics are the basis that determine reservoir diagenesis, pore structure and physical properties. Thin slices of analysis data acquired from 134 core samples in our target formation illustrated that the lithology was mainly composed of feldspar lithic sandstone (Figure 4a). The main composition of debris was metamorphic rocks (9.15%), and the interstitials were dolomite (2.28%) and kaolinite (2.12%) (Figure 4b,c). Formation porosity ranged from 3.6 to 18.0%, and the average porosity was 12.53%. Permeability was distributed from 0.50 to 251.19 mD, and the average permeability was only 7.86 mD (Figure 5). Meanwhile, two interval distributions of permeability revealed complicated pore structures. This resulted in a disordered relationship between porosity and permeability (Figure 1).

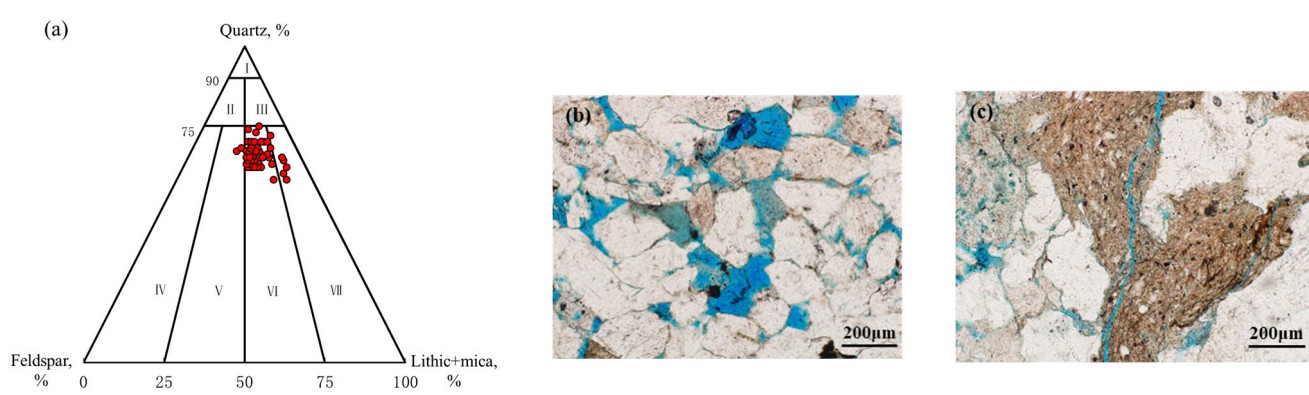

**Figure 4.** Lithology identification based on thin slices analysis data of 134 core samples (**a**), facies type (**b**,**c**) in Pinghu Formation of KQT Region, East China Sea.

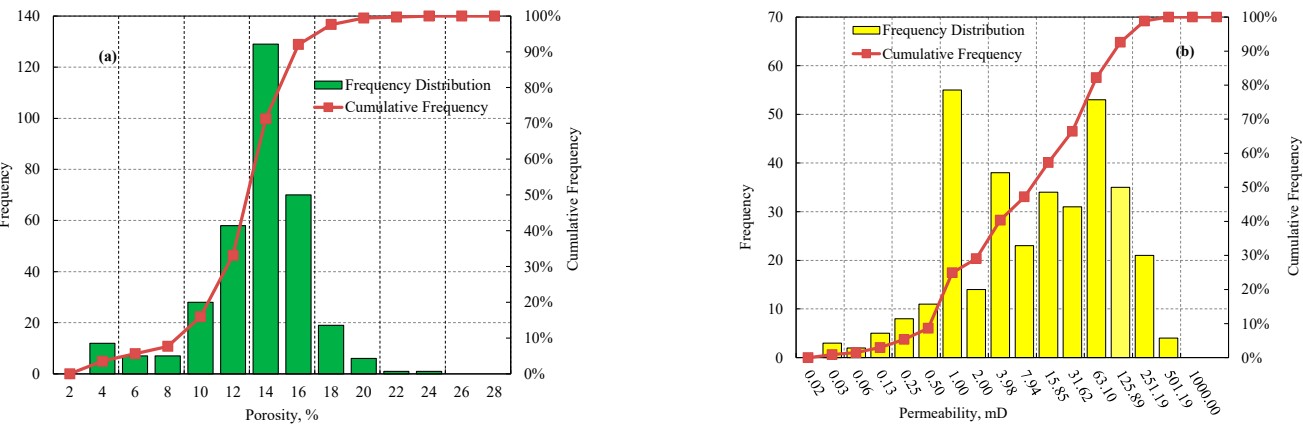

**Figure 5.** Histograms of core-derived porosity (**a**) and permeability (**b**) in Pinghu Formation of KQT Region, East China Sea.

## 4. Method of Pore Structure Characterization and Evaluation

### 4.1. Formation Pore Structure

To quantize formation pore structure, 34 core samples were recovered from Pinghu Formation and applied in mercury injection capillary pressure (MICP) experiments. The measured MICP and corresponded J function curves are displayed separately in Figure 6a,b. During the MICP experiment, the used maximal mercury injection pressure reached 182.04 MPa: this ensured that the whole pore throat size was well exhibited. The threshold pressure of the Pinghu Formation ranged from 0.03 to 1.0 MPa; the maximal pore throat ra-

dius was distributed from 0.15 to 33.78 μm; the median pore throat radius was distributed from 0.015 to 13.07 μm; and the pore structure dominated permeability (Figure 7). To effectively indicate high-quality reservoirs, precisely calculate formation permeability and predict deliverability, formation pore structure should first be characterized.

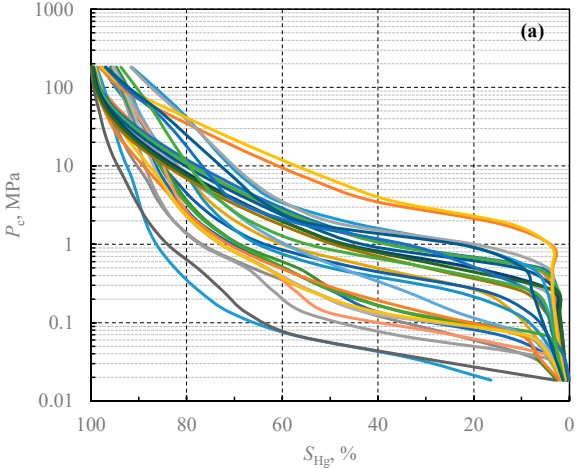
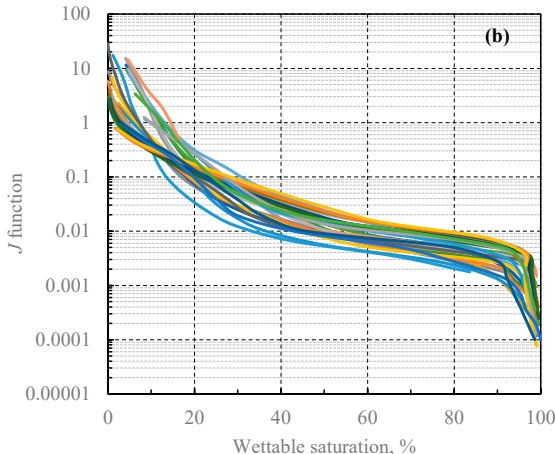

**Figure 6.** MICP and corresponding J function curves acquired from 34 core samples. These core samples were drilled from the Pinghu Formation in KQT Region, East China Sea. In this figure, $P_c$ represents mercury injection pressure in MPa and $S_{Hg}$ represents for mercury injection saturation under every mercury injection pressure in %. J function can be used to express the pore structure and formation type after the effect of physical properties of rock was removed. The shape and position of MICP and J function curves were reflected in rock pore structure. The MICP curve located in the bottom once rock pore structure was good, because mercury can be easily injected into the pore space under the same $P_c$. On the contrary, MICP curve located in the top for rock contains poor pore structure.

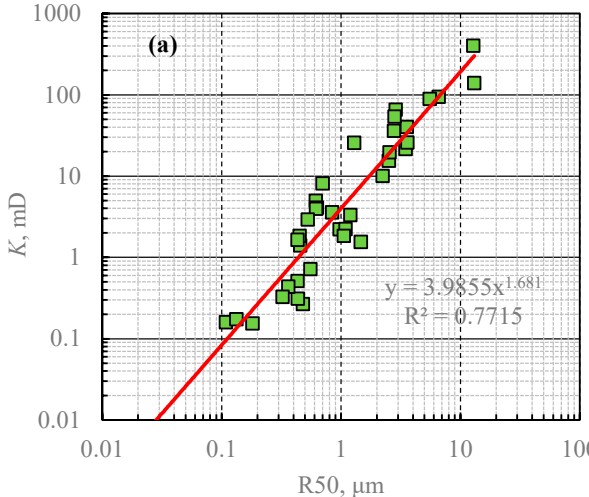
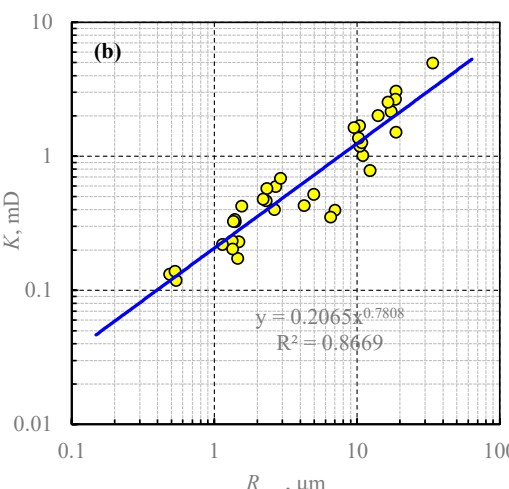

**Figure 7.** Relationships between median pore throat radius (R50) versus permeability (**a**) and maximal pore throat radius ($R_{max}$) and permeability (**b**). These two figures indicate that pore structure was the main factor that controlled permeability in the Pinghu Formation. With the increased of R50 and $R_{max}$ (reflecting formation pore structure), formation permeability increased.

In this study, we improved the pore structure characterization method that was raised by Xiao et al. (2016). Based on NMR and MICP experimental data of 34 core samples, we established the transformation model between $T_2$ time and $P_c$ based on the piecewise function calibration (PFC) method [20]. By using the PFC method, NMR $T_2$ distributions

were transformed as pseudo-$P_c$ curves. Then, low permeability sandstone reservoirs pore structure was characterized in Pinghu Formation.

### 4.2. Theory of Characterizing Pore Structure Based on NMR Logging

Based on NMR theory, NMR $T_2$ relaxation time of a water-wettable rock is dominated by surface relaxation and bulk relaxation; diffusion relaxation can be ignored due to negligible contribution [57]:

$$\frac{1}{T_2} = \rho_2 \left( \frac{S}{V} \right)_{por} = F_S \frac{\rho_2}{r_{por}} \tag{1}$$

where $\rho_2$ is the surface relaxation rate; $S$ is the pore surface area in $\mu m^2$; $V$ is the pore volume in $\mu m^3$; subscript por stands for rock pore size; $r_{por}$ is the pore radius in micrometer; and $F_s$ is the pore shape geometric factor. Its value is constant once pore shape is assumed as regular.

In an air–mercury fluid system, the relationship between capillary pressure and pore throat size can be expressed:

$$P_c = \frac{0.735}{R_c} \tag{2}$$

where $P_c$ is the capillary pressure in MPa; $R_c$ is the pore throat radius in $\mu m$.

Xiao et al. (2016) reported that the relationship between pore size and pore throat radius can be expressed as a power function [20]:

$$r_{por} = p \times (R_c)^q \tag{3}$$

where $p$ and $q$ are the proportionality coefficients that connect pore size with pore throat radius.

Combined with Equations (1)–(3), a derivative formula that connects pore throat radius with $T_2$ relaxation time can be expressed:

$$p \times \left( \frac{0.735}{P_c} \right)^q \approx \rho_2 \times T_2 \times F_S \tag{4}$$

Then,

$$P_c = C \times \left( \frac{1}{T_2} \right)^{\frac{1}{q}} \tag{5}$$

where,

$$C = 0.735 \times p^q \times \left( \frac{1}{F_s \times \rho_2} \right)^{\frac{1}{q}} \tag{6}$$

Once $p$ and $q$ were first calibrated, $C$ can be determined. By using Equation (5), the $P_c$ value can be acquired from NMR logging. If we normalize NMR $T_2$ amplitudes and reversely accumulate them based on the principle displayed in Figure 8, a pseudo-$P_c$ curve can be constructed from NMR data.

### 4.3. Constructing Pseudo-P$_c$ Curves from NMR Logging Based on Formation Classification

Our target low permeability sandstone reservoirs were strongly heterogeneous. To make the extracted $P_c$ curves from NMR logging reliable, the PFC method was used. This method covered several procedures:

First, a certain number of core samples was chosen to simultaneously apply for NMR and MICP experiments. NMR $T_2$ distributions and MICP curves were collected as the basic dataset. In the Pinghu formation of KQT Region, 34 core samples were recovered.

Second, core samples were classified into several types based on physical properties. MICP curves and NMR $T_2$ distributions were also classified by using the same criterion. In this study, 34 core samples were classified into three types. The classification criteria are listed in Table 1.

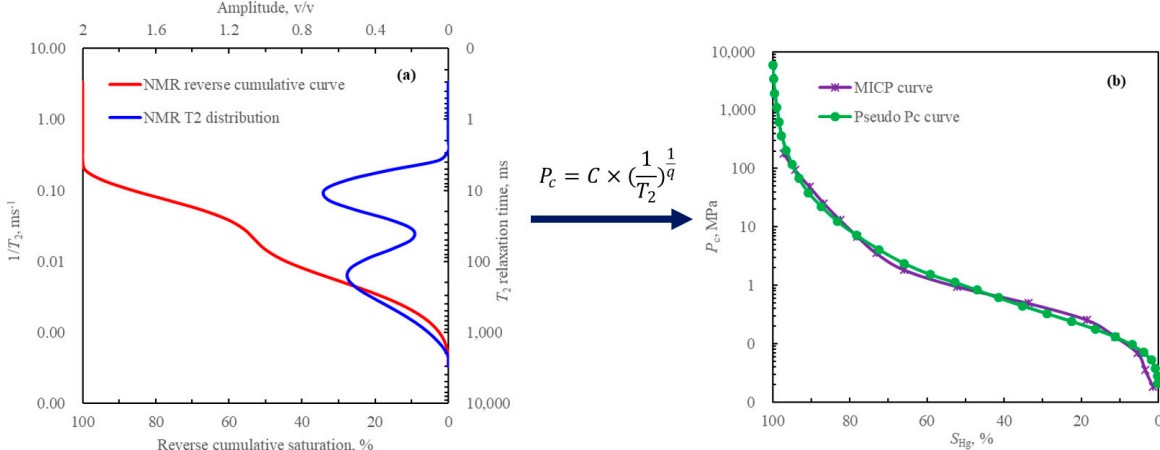

**Figure 8.** Principle of constructing pseudo-$P_c$ curve from NMR data (**a**) and comparison of synthesized pseudo-$P_c$ curve with laboratory measured MICP curve for a core sample (**b**).

**Table 1.** Establishment of rocks classification criteria based on MICP curves and physical properties in the Pinghu Formation of KQT Region.

| Rock Type | Porosity (%) | Permeability (mD) | Median Pore Throat Radius (μm) | Median Pressure (MPa) | Maximum Pore Throat Radius (μm) | R35 (μm) | Threshold Pressure (MPa) | MICP Curve Morphology |
|---|---|---|---|---|---|---|---|---|
| I | 11.3~16.4 | 15.4~402.0 | 1.02~14.57 | 0.05~0.72 | 9.51~33.78 | 2.82~19.69 | 0.02~0.08 | Demarcation of large and small pore throat is obvious |
| II | 10.0~18.9 | 1.41~8.14 | 0.44~1.48 | 0.50~1.67 | 1.36~6.99 | 0.78~2.18 | 0.07~0.49 | Demarcation of large and small pore throat is obvious |
| III | 7.9~10.0 | 0.16~0.44 | 0.11~0.48 | 1.53~6.79 | 0.49~1.45 | 0.22~0.73 | 0.25~1.67 | Demarcation of large and small pore throat isn't obvious |

Third, MICP curves for every type of core sample were processed. Three average MICP curves were obtained. These three MICP curves represented the pore structure of our target low permeability sandstones. Three types of MICP curves and the corresponding three average MICP curves are displayed in Figure 9.

Fourth, NMR $T_2$ spectra were processed by using the same criteria used to obtain three average NMR $T_2$ distributions. These three average NMR $T_2$ distributions are reversibly accumulated and normalized to extract NMR inverse accumulative curves (Figure 10).

Fifth, the transformation model between $T_2$ relaxation time and $P_c$ is established for every type of core sample based on the principle illustrated in Equation (5). It should be noted that the used parameters in the transformation model were different in large pore throats and small pore throats for the same type of core sample. These transformation models were expressed as Equations (7) and (8).

Large pore throat:

$$P_c = C_l \times \left(\frac{1}{T_2}\right)^{\frac{1}{n_l}} \tag{7}$$

Small pore throat:

$$P_c = C_s \times \left(\frac{1}{T_2}\right)^{\frac{1}{n_s}} \tag{8}$$

where $C_l$ and $n_l$ are the parameters involved to transform $T_2$ time as a $P_c$ in the large pore throat; $C_s$ and $n_s$ are the involved parameters to transform $T_2$ time as a $P_c$ value in the small pore throat.

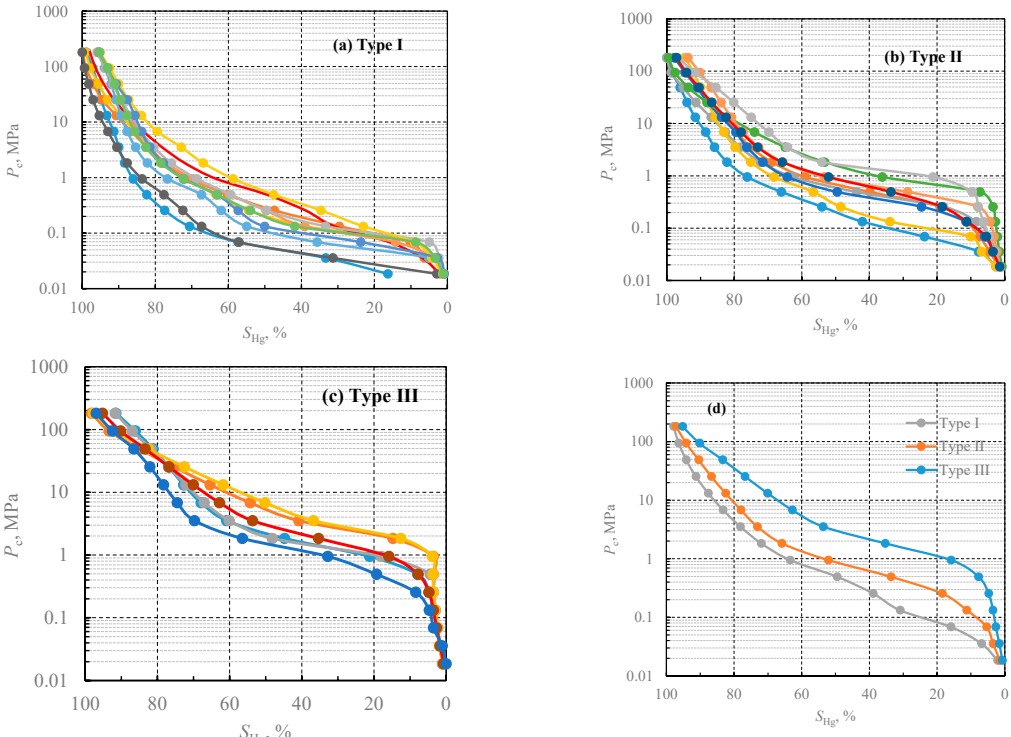

**Figure 9.** MICP curves of three types of core samples (**a**–**c**) and the average MICP curves (**d**) in the Pinghu Formation. Each color in a-c represents a capillary pressure curve that belongs to different formation types. These figures indicated that the first type of formation was dominant in our target formation.

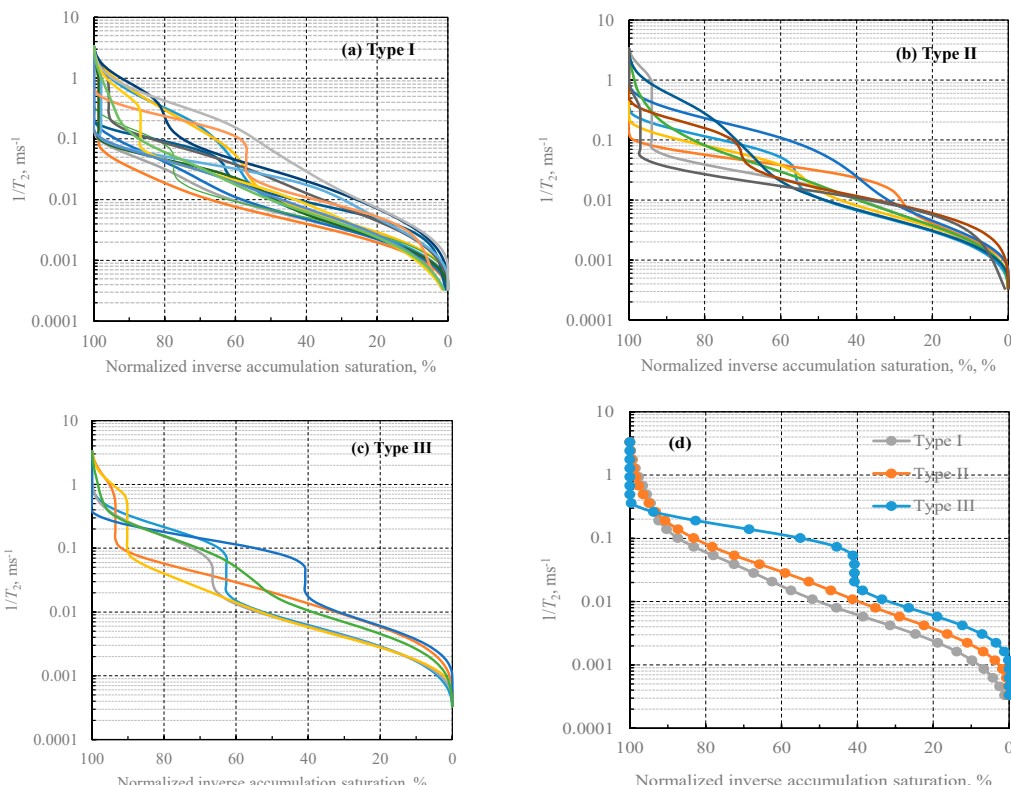

**Figure 10.** Inverse accumulative curves of three types of core samples (**a**–**c**) and the average inverse accumulative curves (**d**) in the Pinghu Formation.

In Figure 11, we exhibited the crossplots of porosity versus permeability and porosity versus R50 for three types of core samples. Obviously, three types of core samples contained differentiated physical characteristics. From the third to the first type of core sample, with the increase of porosity, the corresponding permeability and median pore throat radius also increase; they were clearly separated. This verified the reliability of the listed formation classification criteria in Table 1. Through Figures 9–11, we could notably conclude the difference among these types of formations. The first type of formation contained the best pore structure: it had the lowest threshold pressure and highest median pore throat radius. The corresponding permeability was also high. Potential hydrocarbon productivity of this type of formation was enormous. Pore structure of the second type of formation closely followed. Formation physical property parameters were moderate, and medium to poor pay zone developed. Pore structure and production capacity of the third type of formation were the worst. No effective pay zone can be extracted from such a type of formation.

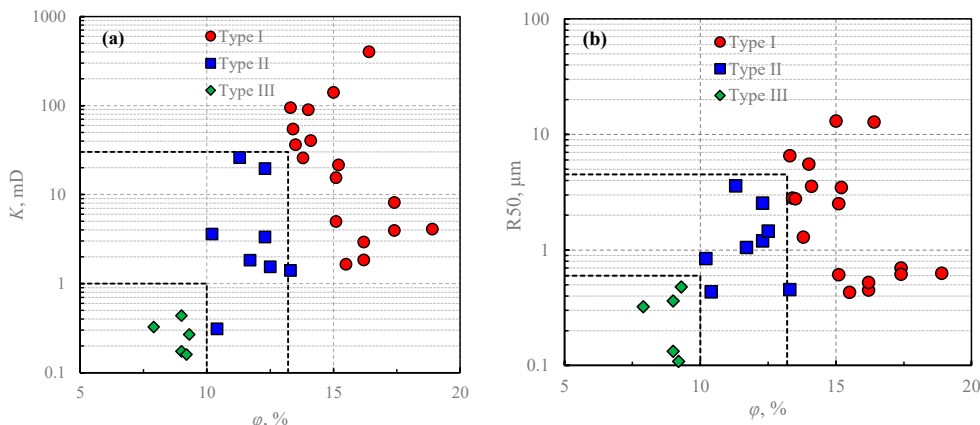

**Figure 11.** Crossplots of porosity versus permeability (**a**) and porosity versus *R*50 (**b**) for three types of core samples in Pinghu Formation.

By combining with Figures 9 and 10, models of constructing pseudo-$P_c$ curves from NMR data were established and displayed in Figure 12. Good power functions existed in two parts for every type of core sample. Once these models were extended into field applications, pseudo-$P_c$ curves can be consecutively synthesized, and they can be used to replace MICP curves to characterize formation pore structure.

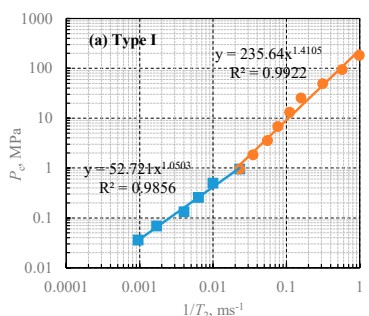 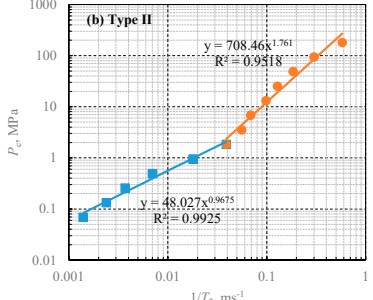 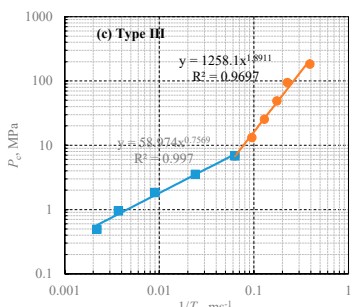

**Figure 12.** Models of transforming NMR $T_2$ distribution as pseudo-$P_c$ curves based on PFC method.

## 5. Estimation of Reservoir Petrophysical Parameters

### 5.1. Porosity Calculation

In gas-bearing formations, porosity cannot be precisely calculated from a single well logging curve due to the excavation effect [58]. Meanwhile, the effect of rock matrix to well logging responses cannot be ignored in low permeability sandstone reservoirs. In the Pinghu Formation, a triangular chart, established based on density and neutron porosities, was used to calculate formation porosity.

The principle of calculating porosity based on triangular chart is shown in Figure 13a. In this method, determining clay point is the key (point B in Figure 13a). Generally, it is determined by using a crossplot of density and neutron (Figure 13b). Afterwards, Equations (9)–(12) were used to calculate density porosity, neutron porosity, clay density porosity and clay neutron porosity, respectively.

$$\varphi_D = \frac{\rho_b - \rho_{ma}}{\rho_f - \rho_{ma}} \tag{9}$$

$$\varphi_N = \text{CNL} + 0.015 \tag{10}$$

$$\varphi_{Dclay} = \frac{\rho_{clay} - \rho_{ma}}{\rho_f - \rho_{ma}} \tag{11}$$

$$\varphi_{Nclay} = \text{CNL}_{clay} + 0.015 \tag{12}$$

where $\varphi_D$ is the porosity calculated from density logging in fraction; $\varphi_N$ is the porosity calculated from neutron logging in fraction; $\varphi_{Dclay}$ is the porosity calculated from density logging in clay point in fraction; $\varphi_{Nclay}$ is the porosity calculated from neutron logging in clay point in fraction; $\rho_b$ is the bulk density, $\rho_{clay}$ is the bulk density in clay point; $\rho_{ma}$ and $\rho_f$ are the bulk density of rock matrix and pore fluid, respectively. The units are is g/cm$^3$; CNL is the neutron logging value; and CNL$_{clay}$ is the neutron logging value in clay point; the unit is $v/v$.

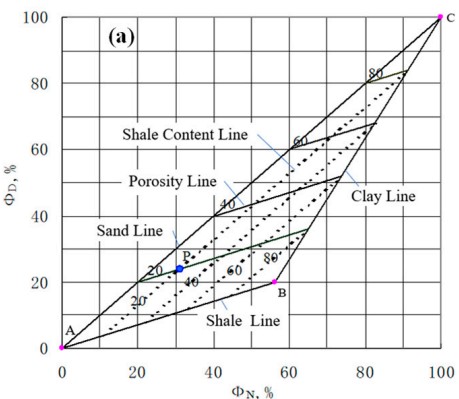 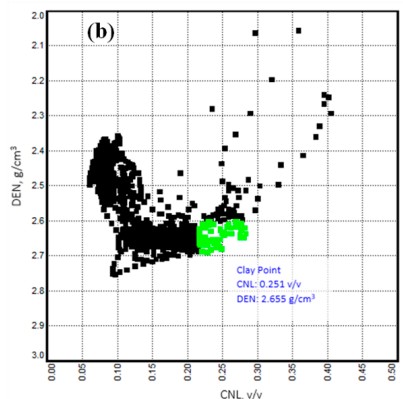

**Figure 13.** Calculation of porosity based on triangular chart of neutron and bulk density (**a**) and determination of clay point based on the crossplot of density and neutron logging (**b**). The green area was the selected clay point.

Combining with $\varphi_D$, $\varphi_N$, $\varphi_{Dclay}$ and $\varphi_{Nclay}$, low permeability sandstone reservoir porosity was calculated based on the triangular chart:

$$\varphi_{ND} = \frac{\left| \varphi_N \times \varphi_{Dclay} - \varphi_D \times \varphi_{Nclay} \right|}{\left| \varphi_{Dclay} - \varphi_{Nclay} \right|} \tag{13}$$

where $\varphi_{ND}$ is the true formation porosity calculated from density and neutron logging in gas-bearing reservoirs.

Meanwhile, the shale content can also be incidentally calculated:

$$V_{sh} = \frac{\varphi_D - \varphi_N}{\varphi_{Dclay} - \varphi_{Nclay}} \tag{14}$$

where *Vsh* is the shale content in *v/v*.

### 5.2. Permeability Calculation

Xiao et al. (2021) pointed out that permeability can be calculated after formation pore structure was first characterized. Several models have been raised [35]. In this study, we attempted to analyze the relationships among permeability, porosity and pore structure parameters based on 34 core samples. Finally, we found that the highest frequency of the pore throat radius statistical histogram was associated with 50% mercury injection saturation (Figure 14). This meant that the biggest contribution to permeability was the median pore throat radius $R50$ (pore throat radius corresponded to 50% mercury saturation) [59,60]. Hence, $R50$ was extracted from 34 core samples, and a model of calculating permeability from $P_c$ curve was established (Figure 15). In this figure, the comprehensive physical property index was defined as the square root of the ratio of permeability and porosity ($\sqrt{K/\varphi}$). Compared with Figure 1, permeability calculation accuracy was improved greatly. Combining with Figures 12 and 15, formation permeability can be calculated well after pore structure was first characterized in the intervals in which field NMR logging was acquired. The permeability prediction model based on $R50$ is expressed:

$$K = \left(0.557 \times R50^{0.7985}\right)^2 \times \varphi = 0.31 \times R50^{1.597} \times \varphi \tag{15}$$

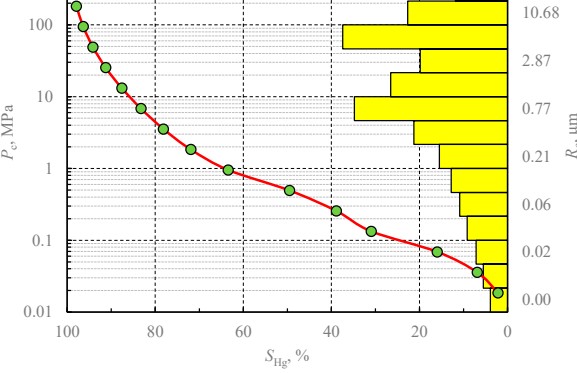

**Figure 14.** MICP curve and pore throat radius distribution for a representative core sample. This figure illustrated that the highest frequency of pore throat radius statistical histogram corresponded to 50% mercury injection saturation. This indicated that R50 was the main factor that controlled rock permeability in the Pinghu Formation.

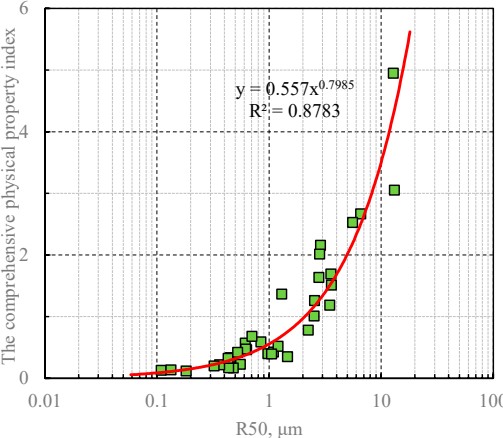

**Figure 15.** Crossplot of median pore throat radius (R50) and comprehensive physical property index for 34 core samples. Permeability could be precisely calculated after formation pore structure was first characterized.

### 5.3. Water saturation Evaluation

In low permeability sandstone reservoirs with strong heterogeneity, no valid model has been proposed. Archie's equation was still used to calculate water saturation (Equations (16)–(18)) [61,62]. To improve low permeability sandstone water saturation estimation, optimizing the involved rock resistivity parameters (cementation exponent *m* and saturation exponent *n*) was an effective approach [38].

$$F = \frac{R_0}{R_w} = \frac{a}{\varphi^m} \tag{16}$$

$$I = \frac{R_t}{R_0} = \frac{b}{S_w^n} \tag{17}$$

Then

$$S_w = \sqrt[n]{\frac{a \times b \times R_w}{\varphi^m \times R_t}} \tag{18}$$

where $\varphi$ is the porosity in $v/v$; $R_0$ is the rock resistivity with fully water saturation; $R_t$ is the rock resistivity with hydrocarbon saturated; and $R_w$ is the formation water resistivity. The unit of the parameters is $\Omega.m$. $S_w$ is the water saturation in $v/v$. $F$ is the formation factor. $I$ is the resistivity index, $m$ is the cementation exponent, $n$ is the saturation exponent, $a$ and $b$ are the coefficient that associated with lithology. $a$, $b$, $m$ and $n$ are collectively referred to as rock resistivity parameters.

### 5.4. Optimization of Cementation Exponent

Figure 2a illustrated that the relationship between porosity and formation factor was not a simple power function, especially in core samples with porosity lower than 8.0%. If we took the logarithm of porosity and formation factor and put $\log_{10}(\varphi)$ and $\log_{10}(F)$ in a linear coordinate, the relationship between porosity and formation factor can be expressed by a quadratic function:

$$\log_{10}(F) = x \times \log_{10}^2(\varphi) + y \times \log_{10}(\varphi) + z \tag{19}$$

where $x$, $y$ and $z$ are the undetermined coefficients.

Considering the boundary condition that the trendline of $\log_{10}(\varphi)$ versus $\log_{10}(F)$ should pass (0.0), the value of $z$ was defined as 0.0.

Combined with Equations (16) and (19), the value of $a$ was found to be 1.0. $m$ can be calculated from porosity using the following equation:

$$m = x \times \log_{10}(\varphi) + y \tag{20}$$

Afterwards

$$F = \frac{1}{\varphi^{x \times \log_{10}(\varphi) + y}} \tag{21}$$

By using the mentioned method in Equation (21), measured data displayed in Figure 2a were reprocessed. A model of calculating cementation exponent from porosity in the Pinghu Formation was acquired in Equation (22). A novel relationship between porosity and formation factor was displayed in Figure 16. Compared with Figure 2a, dependency between these two parameters greatly improved. Meanwhile, various cementation exponents, but not a fixed value, can be extracted to calculate water saturation.

$$m = 0.358 \times \log_{10}(\varphi) + 1.95 \tag{22}$$

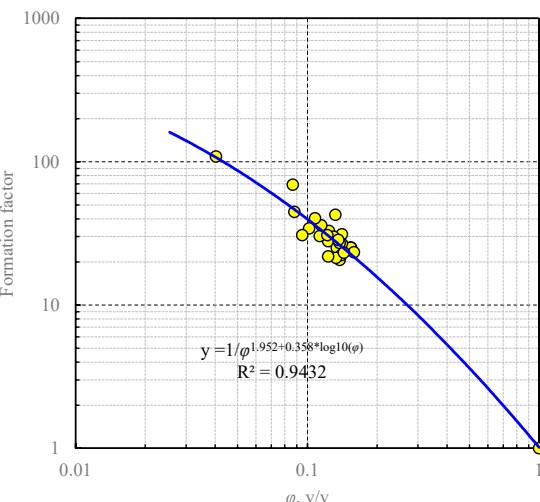

**Figure 16.** Novel relationship between porosity and formation factor in low permeability sandstones in Pinghu Formation. This figure illustrated that the relationship between porosity versus formation factor was well expressed by using various cementation exponent, especially for rocks with porosity lower than 8.0%. If we directly used Archie's equation, cementation exponent would be overestimated and water saturation should be underestimated.

### 5.5. Extraction of Saturation Exponent Based on Formation Classification

To acquire accurate saturation exponents, core samples displayed in Figure 2b were classified into three types by using the criteria listed in Table 1. For each type of core sample, the relationship between water saturation and resistivity index was established. The results are shown in Figure 17. From the first to the third type of rock, the saturation exponent increased. This meant that saturation exponent increased with deterioration of formation pore structure. After core samples were classified into three types, the correlation between water saturation and resistivity index improved greatly. Precise saturation exponents can be obtained.

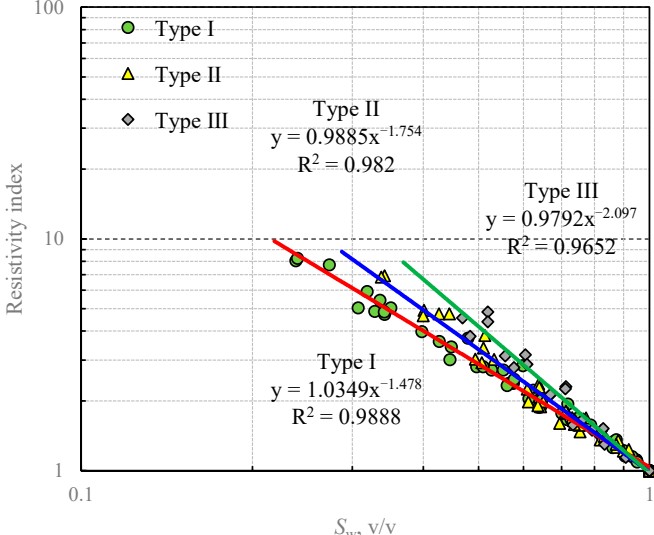

**Figure 17.** Improved relationships between water saturation versus resistivity index in low permeability sandstones in the Pinghu Formation. This figure illustrated that saturation exponent is heavily affected by pore structure. From the first to third type of rocks (green dots, yellow triangles and gray diamonds, respectively), saturation exponents were gradually increasing due to pore structure declining.

### 5.6. Estimation of Irreducible Water Saturation ($S_{wirr}$)

Irreducible water saturation ($S_{wirr}$) is an important parameter in evaluation of low permeability sandstone reservoirs and identification of pore fluids, because high $S_{wirr}$ is a key factor that causes contrast of low resistivity in such type of reservoirs [3]. Generally, $S_{wirr}$ is calculated from NMR logging by using a $T_{2cutoff}$. However, since a reasonable $T_{2cutoff}$ cannot be consecutively acquired in a whole well, a fixed $T_{2cutoff}$ of 33 millisecond (ms) was always used [63]. In the Pinghu Formation, the experimental values of $T_{2cutoff}$s were divergent (Figure 18). It was difficult to calculate $S_{wirr}$ based on the method related to the $T_{2cutoff}$. To establish a model to calculate $S_{wirr}$, relationships among $S_{wirr}$ with other formation parameters were analyzed. Finally, we found that $S_{wirr}$ was heavily associated with comprehensive physical property index (Figure 19). Hence, $S_{wirr}$ can be calculated once porosity and permeability were available, even if no NMR logging data were acquired.

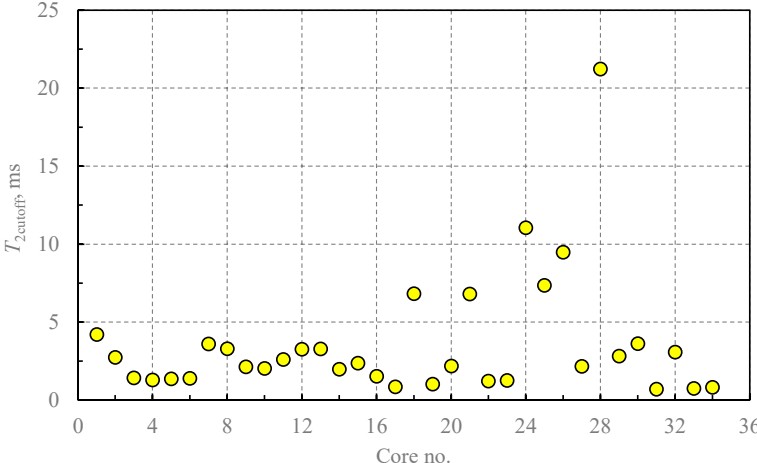

**Figure 18.** Experimental NMR $T_{2cutoff}$s of core samples in the Pinghu Formation. The measured $T_{2cutoff}$s were not a fixed value; this resulted in the difficulty of the $S_{wirr}$ calculation.

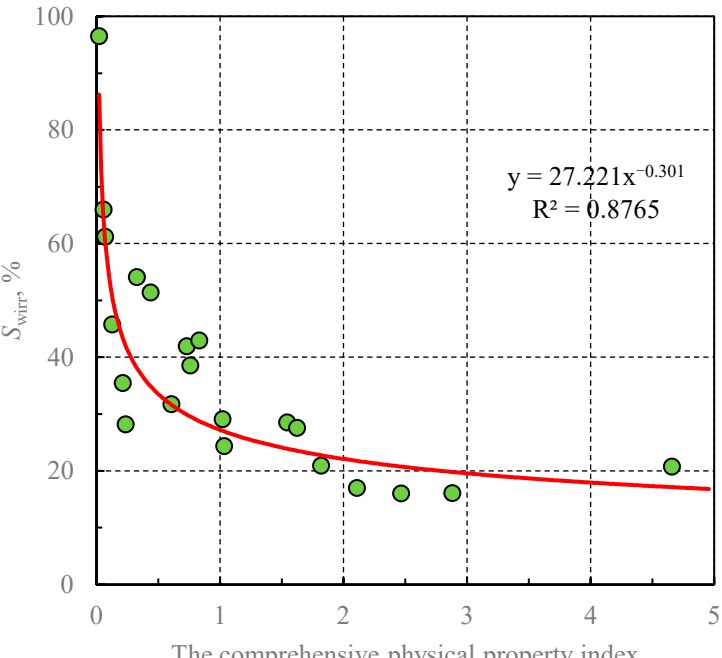

**Figure 19.** Relationship between $S_{wirr}$ and comprehensive physical property index in low permeability sandstones in the Pinghu Formation. With the formation pore structure becoming better, the corresponding $S_{wirr}$ decreased.

## 6. Identification of Pore Fluids Based on Geophysical Well Logging

### 6.1. Identifying Pore Fluids Based on Apparent Formation Water Resistivity

Generally, pore fluids are always identified based on resistivity and many crossplots, e.g., the crossplot of porosity versus resistivity, and water saturation versus resistivity, were raised [64]. However, resistivity cannot be directly used to identify pore fluids due to a contrast of low resistivity in low permeability sandstones [55]. In this study, we raised a method, named the improved apparent formation water resistivity, to identify pore fluids.

For rocks fully saturated with water, the relationship among $R_0$, $R_w$ and porosity is expressed in Equation (16). Combined with Equations (16) and (20) and after some transformation, a derivative equation can be written as

$$R_w = R_0 \times \varphi^{x \times log_{10}(\varphi)+y} \tag{23}$$

If we displace $R_0$ by $R_t$ in Equation (23), formation water resistivity can still be calculated. However, it was not true formation water resistivity. Thus, we named it as apparent formation water resistivity and defined it as $R_{wa}$:

$$R_{wa} = R_t \times \varphi^{x \times log_{10}(\varphi)+y} \tag{24}$$

where $R_{wa}$ is the apparent formation water resistivity in $\Omega.m$.

Generally, $R_{wa}$ is equal to $R_w$ in water saturated layers, because measured $R_t$ was equal to $R_0$. However, in hydrocarbon-bearing reservoirs, $R_t$ deviated from $R_0$ and its value varied due to the effects of many factors. These factors included porosity, content of saturated hydrocarbon, pore structures, and so on. Thus, the calculated $R_{wa}$ was not a fixed value in a whole interval, but rather, fluctuated around a certain value as a normal distribution. In hydrocarbon-bearing reservoirs, the distribution range of $R_{wa}$ was wide. On the contrary, $R_{wa}$ was narrowly distributed in water saturated layers. The distribution range was situated between these two in hydrocarbon and water formation (Figure 20a,c).

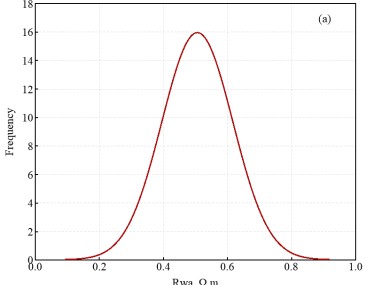 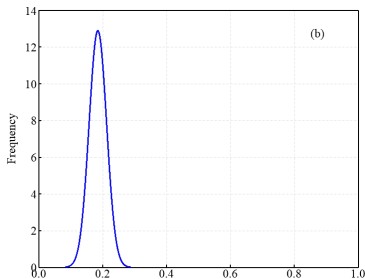 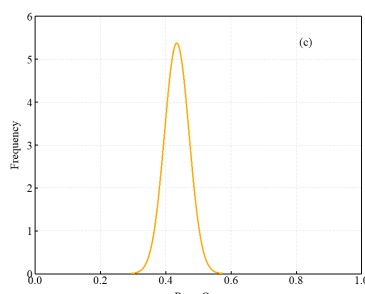

**Figure 20.** Distributions of $R_{wa}$ in formation with different pore fluids. Based on the morphology of $R_{wa}$ distribution, pore fluids can be qualitatively identified. Hydrocarbon-bearing reservoirs contained wide $R_{wa}$ distribution (**a**), $R_{wa}$ distribution of water saturated layers was narrow (**b**). Hydrocarbon and water formation is situated between these two (**c**).

To quantitatively identify pore fluids by using $R_{wa}$ distribution, we extracted two parameters: $R_{wa}$ mean value and $R_{wa}$ variance. These two parameters, respectively, represented the position and width of $R_{wa}$ distribution in an interval and are expressed as follows:

$$R_{wam} = \frac{\sum_{i=1}^{k} R_{wa}(i) \times amp(i)}{\sum_{i=1}^{k} amp(i)} \tag{25}$$

$$R_{wav} = \sqrt{\frac{\sum_{i=1}^{k} amp(i) \times (R_{wa}(i) - R_{wam})^2}{\sum_{i=1}^{k} amp(i)}} \tag{26}$$

where $R_{wam}$ is the mean value of apparent formation water resistivity, $R_{wav}$ is the variance of apparent formation water resistivity, $R_{wa}(i)$ is the $i^{th}$ $R_{wa}$, and the unit of them is $\Omega.m$.

Amp($i$) is the amplitude that corresponds to $R_{wa}(i)$, $k$ is the number of calculated $R_{wa}$ in a whole interval.

By using Equations (25) and (26), we processed 11 intervals that contained drill stem test (DST) data in the Pinghu Formation in the KQT Region. $R_{wam}$ and $R_{wav}$ were, respectively, calculated. We found that the crossplot of these two parameters was very effective in distinguishing hydrocarbon-bearing reservoirs from water saturated layers (Figure 21). This figure indicated that hydrocarbon-bearing reservoirs always contain high $R_{wam}$ and $R_{wav}$, whereas values of these two parameters were low in water saturated layers. Criteria of identifying pore fluids based on $R_{wam}$ and $R_{wav}$ are listed in Table 2.

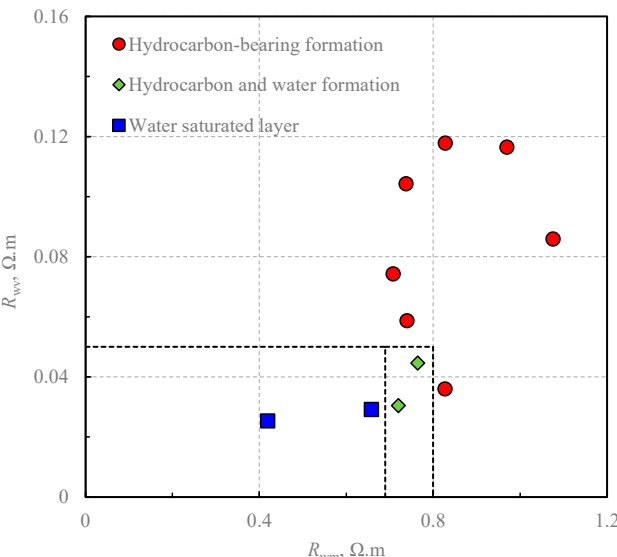

**Figure 21.** Identification of pore fluids based on crossplot of mean value of apparent formation water resistivity ($R_{wm}$) and variance of apparent formation water resistivity ($R_{wv}$).

**Table 2.** Criteria of identifying pore fluids based on two methods raised in this study.

| Pore Fluid | $R_{wam}$ (Ω.m) | $R_{wav}$ (Ω.m) | $S_w$ (%) | $S_{wf}$ (%) |
|---|---|---|---|---|
| Hydrocarbon-bearing formation | Greater than 0.80 | Greater than 0.05 | Less than 60.00 | Less than 27.00 |
| Hydrocarbon and water formation | 0.69~0.80 | Lower than 0.05 | 60.00~70.50 | 27.00~60.00 |
| Water saturated layer | Lower than 0.69 | Lower than 0.05 | Greater than 70.50 | Greater than 60.00 |

### 6.2. Identifying Pore Fluids Based on $S_w$ and $S_{wirr}$

In addition to above-mentioned method, we also proposed the second method, named the overlapping method of $S_w$ *and* $S_{wirr}$, to identify pore fluids. In hydrocarbon-bearing reservoirs, hydrocarbon occupies the big pore space; irreducible water is adsorbed on the pore surface and occupies in the small pore space; and there was no movable water. Hence, $S_{wirr}$ was infinitely close to $S_w$. On the contrary, besides irreducible water, abundant movable water was present in water saturated layers. Hence, we raised a parameter of free water saturation ($S_{wf}$), which was defined as the difference of $S_w$ and $S_{wirr}$, to characterize movable water content. We established a crossplot of $S_w$ versus $S_{wf}$ to indicate pore fluids (Figure 22). Based on this crossplot, we raised the criteria of identifying pore fluids and listed them in Table 2.

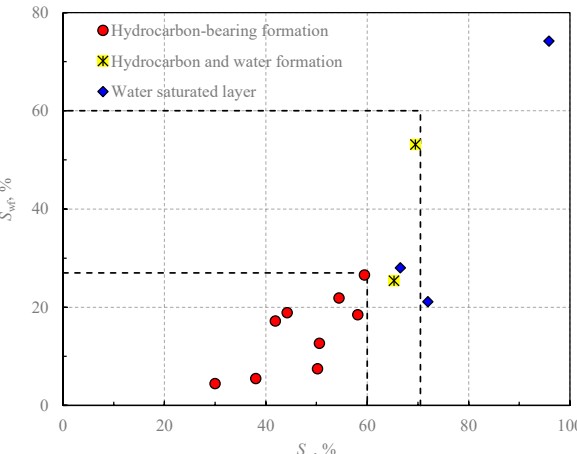

**Figure 22.** Identification of pore fluids based on crossplot of $S_w$ and movable water saturation ($S_{wf}$) in the Pinghu Formation.

## 7. Field Applications

Based on the proposed technique and models, five wells in the KQT Region were processed. Figure 23 displays a field example of processing and interpreting low permeability sandstone reservoirs based on conventional and NMR logging data. In the first track, we displayed the natural gamma ray (GR), spontaneous potential (SPSD) and caliper (CAL) curves; they were used to identify effective sandstone formations. Bulk density (DEN), neutron (CNL) and interval transit time (DT) were displayed in the second track; they were usable in porosity calculation. Deep laterolog (RD) and shallow laterolog (RS) resistivities were exhibited in the third track, and they reflected formation electrical properties. In the fourth track, AMP_DIST was NMR $T_2$ spectrum, which was acquired from Halliburton's MRIL-Prime tool. In the fifth track, we compared constructed pseudo-$P_c$ curves (PC_DIST exhibited as variable density) and measured laboratory capillary pressure curves of core samples (red line). It should be noted that the exhibited pseudo-$P_c$ curves and experimental capillary pressures had been transformed from an air–mercury system to an oil–water system to make them much reasonable in reflecting in-suit formation pore structure. The transformation formula is expressed as Equation (27).

$$(P_c)_{w\_o} = (P_c)_{air\_Hg} \times \frac{\sigma_{w\_o} cos\theta_{w\_o}}{\sigma_{air\_Hg} cos\theta_{air\_Hg}} \tag{27}$$

where $(P_c)_{w\_o}$ and $(P_c)_{air\_Hg}$ are, respectively, the capillary pressure in water–oil and air–mercury systems in MPa. $\sigma_{w\_o}$ and $\sigma_{air\_Hg}$ are the surface tension between two phases of fluids in water-oil and air-mercury systems in dyn/cm, respectively, whereas $\theta_{w\_o}$ and $\theta_{air\_Hg}$ are the contact angles between the two phases of fluids in ($^O$).

Good consistency of constructed pseudo-$P_c$ curves and experimental results in laboratory indicated the reliability of characterizing pore structure by using synthesized $P_c$ curves. RC_DIST displayed in the sixth track was pore throat radius distribution extracted from the pseudo-$P_c$ curve. In the seventh and eighth tracks, we compared calculated porosity (PHIT) and permeability (PERM) with core-derived results. Meanwhile, median pore throat radius (R50) and maximal pore throat radius (RMAX), extracted from pseudo-$P_c$ curves, were compared with core-derived results in the ninth and tenth tracks. Good consistency between calculated parameters by using the proposed technique and core-derived results illustrated the value of raised methods. In the last track, we compared water saturation calculated by using the improved Archie's equation (SWC) and irreducible water saturation (SWIRR). Combined with constructed $P_c$ curves, pore throat radius distribution and pore structure parameters, we can conclude that the intervals of ×187.00 to ×202.50 m and ×231.30 to ×248.60 m were high-quality formations with good pore structure and superior pore throat connectivity. Meanwhile, overlapping of SWC and SWIRR illustrated that

the upper formation contained ultra-low water saturation, and no movable water existed. Hence, this interval was identified as hydrocarbon-bearing reservoirs. However, although the interval of ×231.30 to ×248.60 m contained relatively high resistivity and good pore structure, SWC and SWIRR curves were separated, and SWC was higher than 60.0%. This meant that the pore spaces contained plenty of movable water; it was considered as a pure water saturated layer. In these two intervals, $R_{wam}$ and $R_{wav}$ also indicated that the pore fluid of the upper layer was hydrocarbon, whereas the lower formation was water (Figure 24). These interpretation results were verified by DST data. The DST data, which were acquired from the interval of ×186.7 to ×202.4 m, indicated that approximated 338.5 bbl of oil and 22.87 × $10^4$ m$^3$ of gas were produced per day with no water. However, in the interval of ×231.70 to ×244.00 m, approximately 48.15 m$^3$ of water was produced per day. This verified the reliability of the proposed methods. If we only observed resistivity curves, this interval was easily misjudged as a hydrocarbon-bearing reservoir.

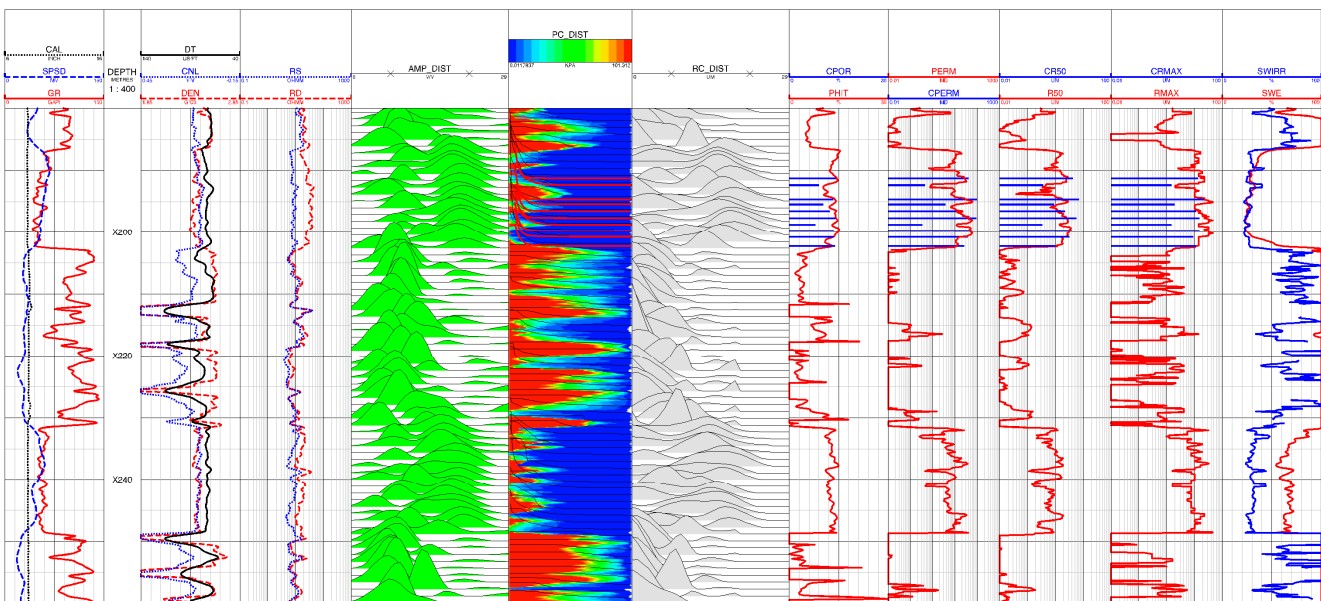

**Figure 23.** A field example of characterizing formation pore structure, calculating formation parameters and identifying pore fluids based on the proposed techniques in this study. By using our raised techniques, the upper layer was identified as a hydrocarbon-bearing formation, and the lower interval was a water saturated layer. If we only used resistivity curves, the lower formation would be misidentified.

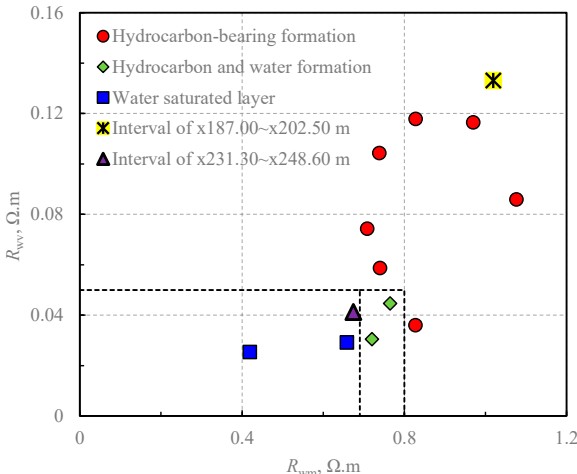

**Figure 24.** Pore fluids identification results of two intervals in a well displayed in Figure 23.

## 8. Extensive Application

To further evaluate the applicability of the proposed methods, they were applied in HG Formation of NB Region, East China Sea to characterize formation pore structure and identify pore fluids. The difference was that HG Formation was divided into four categories to calibrate involved model parameters in Equations (7) and (8) because the pore structure of Huagang Member was very complicated. Figure 25 exhibits models of transforming NMR $T_2$ distribution as pseudo-$P_c$ curves for four types of formations. Good relations still exist for every type of formation. This ensures that formation pore structure can be characterized well. Afterwards, formation porosity, permeability, water saturation ($S_w$) and irreducible water saturation ($S_{wirr}$) can also be calculated.

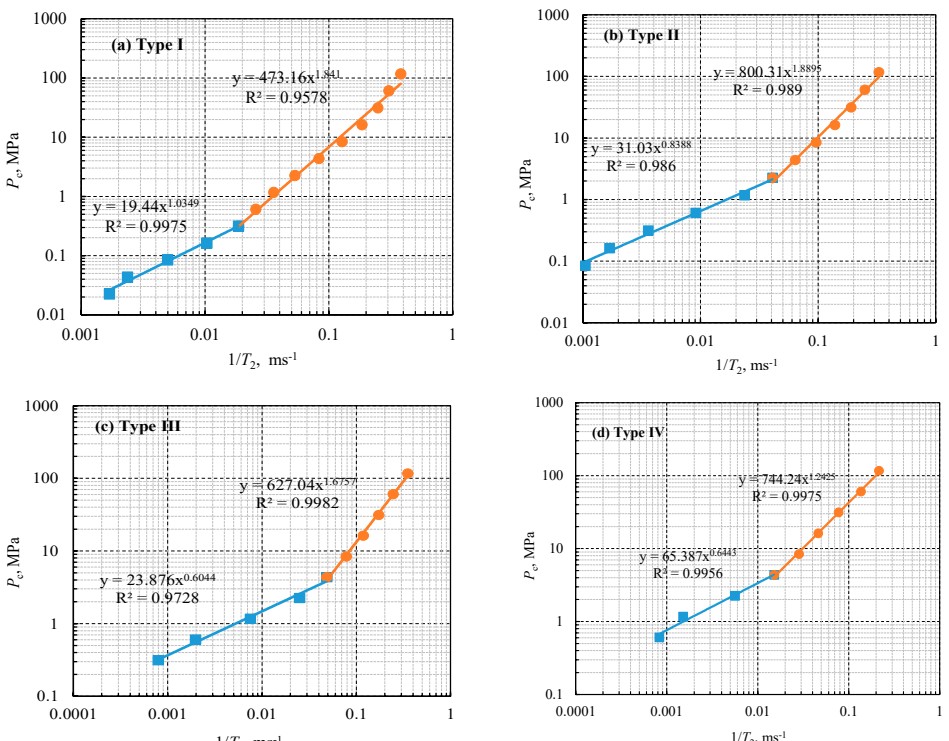

**Figure 25.** Models of transforming NMR $T_2$ distribution as pseudo-$P_c$ curves based on PFC method in HG Formation of NB Region, East China Sea.

In addition, we also established standards to identify hydrocarbon-bearing reservoirs based on apparent formation water resistivity and two water saturation overlap methods. Crossplots of $R_{wam}$ and $R_{wav}$ and $S_w$ versus $S_{wf}$ were raised and shown in Figure 26a,b, separately. High compliance demonstrated the superiority of the proposed method.

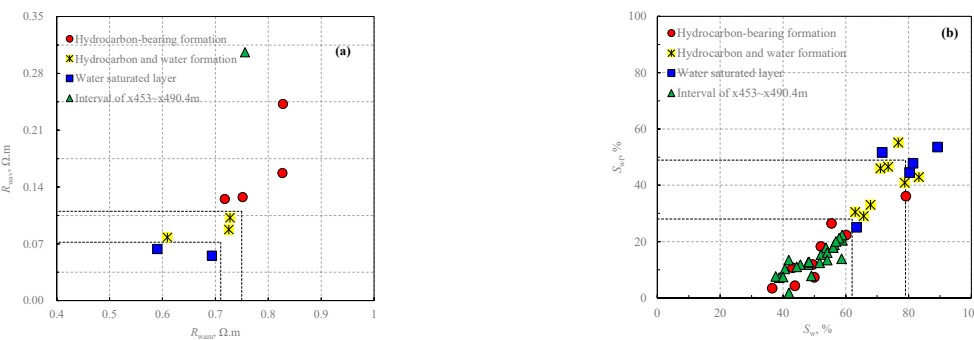

**Figure 26.** Crossplots of $R_{wam}$ versus $R_{wav}$ (**a**) and $S_w$ versus $S_{wf}$ (**b**) that used to identify pore fluids in HG Formation of NB Region, East China Sea.

In Figure 27, we exhibited a field example of characterizing formation pore structure and identifying pore fluids in the HG Formation of the NB Region. Observing constructed pseudo-$P_c$ curves and pore throat radius distributions, an interval of ×455.3 to ×490.4 m was considered as high-quality formations, with the exception of some interbeds with high gamma rays. Based on the standards established in Figure 26a,b, this interval was identified as a hydrocarbon-bearing formation. This identification was verified by DST data acquired in the interval of ×462.0 to ×491.0 m, which showed that approximately $9.51 \times 10^4$ m$^3$ of gas was produced per day.

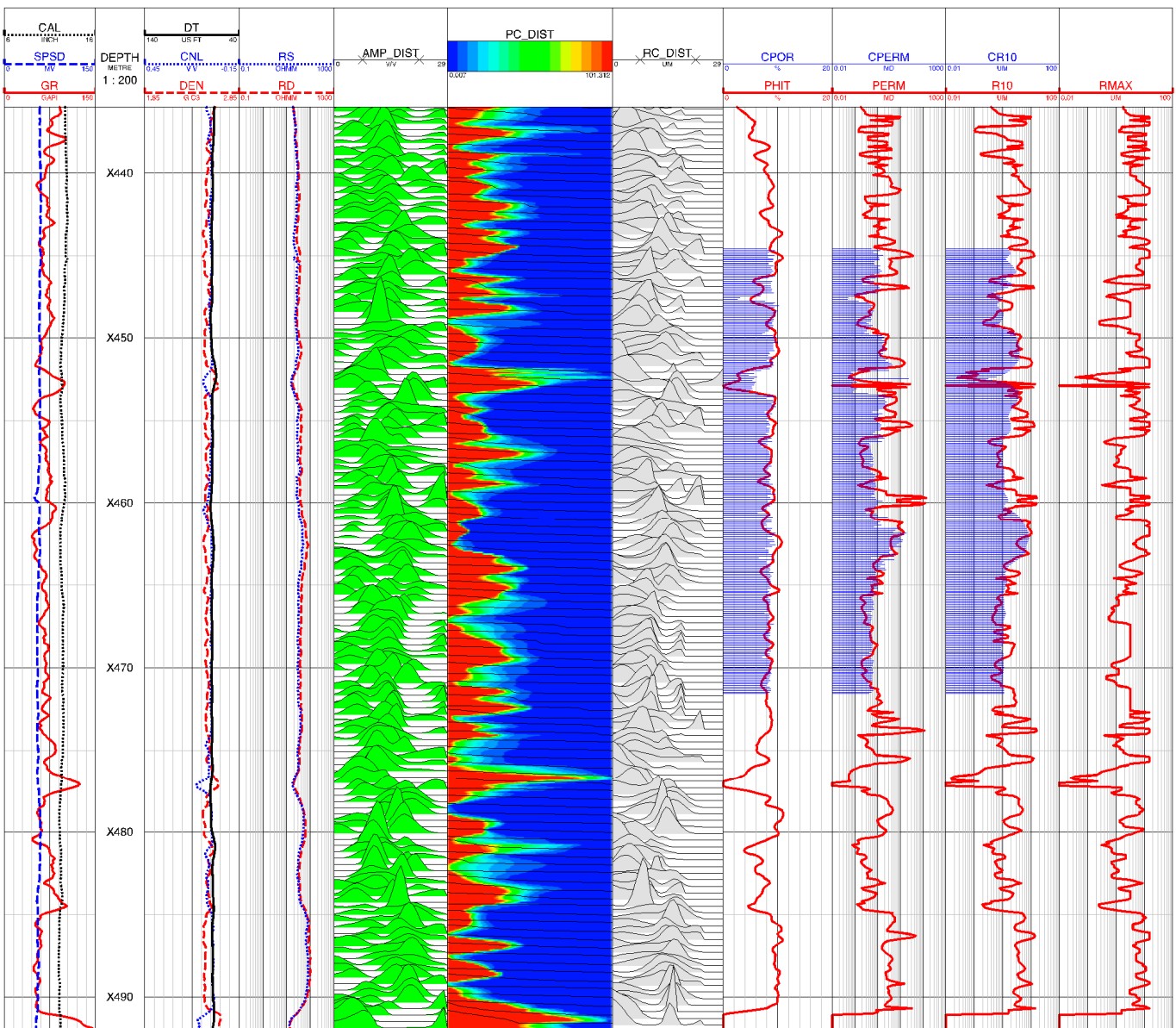

**Figure 27.** A field example of characterizing formation pore structure, calculating formation parameters and identifying pore fluids in HG Formation of NB Region.

## 9. Conclusions

Complicated pore structure and strong heterogeneity limited formation evaluation and characterization based on conventional well logging data. After formation pore structure was quantitatively characterized from NMR logging, reasonable methods and techniques were raised to predict petrophysical parameters and identify pore fluids. Several conclusions can be summarized as follows:

1. Combined with NMR and capillary pressure theories, a method which can be used to transform NMR $T_2$ spectrum as pseudo-$P_c$ curve was established. The method was named the piecewise function calibration method. It was used to quantitatively characterize low permeability sandstone reservoir pore structure and classify reservoirs into three categories in the Pinghu Formation.

2. The triangular chart of neutron and density was used to well calculate porosity. A model which uses median pore throat radius as an input parameter was introduced to estimate permeability from pseudo-$P_c$ curve. For the water saturation calculation, Archie's equation was used, and the involved rock resistivity parameters were optimized. Field examples illustrated that the proposed methods are valuable in our target Pinghu Member.

3. Two techniques, used to identify pore fluids based on the crossplots of mean value of apparent formation water resistivity versus variance of apparent formation water resistivity; and water saturation versus irreducible water saturation, were raised. Field examples in two different regions illustrated that these techniques were valuable in indicating pore fluids. They can be widely used in low permeability sandstones with similar physical properties, whereas common methods would lose their role due to low resistivity contrast. Our raised methods and techniques can further improve complicated formation characterization and allow for high-quality reservoir predictions.

**Author Contributions:** L.X. conceived the idea and F.G. wrote the initial draft of the paper. Z.Z. and E.Y. contributed to carrying out core experiments. W.Z. and W.C. performed the statistical analysis and editing part of the paper. All authors have read and agreed to the published version of the manuscript.

**Funding:** This research was supported by the National Natural Science Foundation of China (No. 41302106), the China Postdoctoral Science Foundation (No. 2012M520347, 2013T60147), and the Fundamental Research Funds for the Central Universities, China (No. 2-9-2016-007).

**Data Availability Statement:** Not applicable.

**Conflicts of Interest:** The authors declare no conflict of interest.

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
