# Peer review of "Low Permeability Gas-Bearing Sandstone Reservoirs Characterization from Geophysical Well Logging Data: A Case Study of Pinghu Formation in KQT Region, East China Sea"

_processes, doi:10.3390/pr11041030_

Round 1
Reviewer 1 Report
General comments
Research based on a robust dataset, I can see several meters logged, multiple techniques used (including NMR!), and high values of R2 in the petro-physical graphs. The manuscript also fits the scope of either the journal, and the specific section (paper on geo-energy in a section on energy systems; the authors study a reservoir system). The conclusions are supported by the data, but the authors need to focus on the organization of the text, and the language.
Specific comments
Line 11-12. “Pinghu Formation in KQT Region, East China Sea belonged to typical low permeability sandstone reservoir” sentence unclear. I suggest “The Pinghu Formation represents a low permeability sandstone reservoir in the KQT Region, East China Sea”
Line 24. “Calculated formation physical property parameters” change into “Computed petrophysical parameters”
Line 27. “Low permeability” and “sandstone reservoirs” are two different key words
Lines 62-64. “Permeability calculation always faced great challenges in low permeability reservoirs due to complicated pore structure and strong heterogeneity, and thus led to poor relationship between porosity (φ) and permeability (K)”. Add recent and relevant review papers on the strong impact on fluid flow of heterogeneities in sandstones, see below:
- Tellam, J.H. and Barker, R.D., 2006. Towards prediction of saturated-zone pollutant movement in groundwaters in fractured permeable-matrix aquifers: the case of the UK Permo-Triassic sandstones. Geological Society, London, Special Publications, 263(1), pp.1-48.
- Medici, G. and West, L.J., 2022. Review of groundwater flow and contaminant transport modelling approaches for the Sherwood Sandstone aquifer, UK; insights from analogous successions worldwide. Quarterly Journal of Engineering Geology and Hydrogeology, 55(4), https://doi.org/10.1144/qjegh2021-176
Line 140. Clearly state the principal aim of your research, and the three to four specific objectives by using numbers (e.g., i, ii, iii). This amendment refers to the final part of your introduction.
Lines 147-149. “During Paleocene-Eocene rifting period, it developed contemporaneous faults and a lot of structural traps, and it became the group of fault blocks that have fallen steadily along the trend”. Please, revise the language
Line 150. Specify the depositional palaeo-environment of the Pinghu Formation. Fluvial, deltaic, or turbiditic sandstones?
Lines 170-412. There is need of a specific methodology section?
Line 412. “Case study”, the title is unclear. I suggest to change the name
Line 476. Add a couple of summary sentences before the bulleting points of your conclusions
Line 494. Add a sentence that summarizes the “take home message” of your research
Line 510. Add the relevant references suggested above
Figures and tables
Figures 22 and 23. Please, combine the two figures together
Author Response
1) Line 11-12. “Pinghu Formation in KQT Region, East China Sea belonged to typical low permeability sandstone reservoir” sentence unclear. I suggest “The Pinghu Formation represents a low permeability sandstone reservoir in the KQT Region, East China Sea”
Response: Done.
2) Line 24. “Calculated formation physical property parameters” change into “Computed petrophysical parameters”.
Response: Done.
3) Line 27. “Low permeability” and “sandstone reservoirs” are two different key words.
Response: Done.
4) Lines 62-64. “Permeability calculation always faced great challenges in low permeability reservoirs due to complicated pore structure and strong heterogeneity, and thus led to poor relationship between porosity (φ) and permeability (K)”. Add recent and relevant review papers on the strong impact on fluid flow of heterogeneities in sandstones, see below:
Response: Thanks for your comments, we have added three recently published references.
5) Line 140. Clearly state the principal aim of your research, and the three to four specific objectives by using numbers (e.g., i, ii, iii). This amendment refers to the final part of your introduction.
Response: Done.
6) Lines 147-149. “During Paleocene-Eocene rifting period, it developed contemporaneous faults and a lot of structural traps, and it became the group of fault blocks that have fallen steadily along the trend”. Please, revise the language.
Response: Thanks for your comments, we have revised the language.
7) Line 150. Specify the depositional palaeo-environment of the Pinghu Formation. Fluvial, deltaic, or turbiditic sandstones?
Response: Done
8) Lines 170-412. There is need of a specific methodology section?
Response: Done
9) Line 412. “Case study”, the title is unclear. I suggest to change the name.
Response: Done
10) Line 476. Add a couple of summary sentences before the bulleting points of your conclusions.
Response: Done
11) Line 494. Add a sentence that summarizes the “take home message” of your research.
Response: Done
12) Line 510. Add the relevant references suggested above.
Response: Done
Reviewer 2 Report
This work proposed a method to evaluate the parameters of low permeability sandstone reservoirs using NMR logging. The method and result should be useful for actual reservoir evaluation. The manuscript should be accepted for publication after a minor revision.
1)Some of the formatting is wrong, for example, the pore structure in the keywords should be followed by a comma. The "Type II" annotation in figure 25 is missing. There are "dots" after some names in the reference. Please check the full manuscript carefully.
2)The font size of all figures in the manuscript should be the same. For example, the font sizes of the three images in figure 3 are obviously not uniform.
3)What is the meaning of "Fig.12" before line 463?
4)he resolution of figures 23 and 27 needs to be improved. As you can see from the figures, some curves have wrong units.
5)The low porosity and low permeability reservoir have been studied in the following literature, which may have some implications on this study.
Xiao K, Duan ZY, Yang YX, et al. Experimental study of relationship among acoustic wave, resistivity and fluid saturation in coalbed methane reservoir. Acta Geophysica,2022.
Author Response
1) Some of the formatting is wrong, for example, the pore structure in the keywords should be followed by a comma. The "Type II" annotation in figure 25 is missing. There are "dots" after some names in the reference. Please check the full manuscript carefully.
Responses: Thanks for your good comments, we have carefully checked the revised manuscript, many mistakes have removed. "Type II" in Fig. 25 is added.
2) The font size of all figures in the manuscript should be the same. For example, the font sizes of the three images in figure 3 are obviously not uniform.
Responses: We have unified the font in all figures that can be edited . However, in some directly cited figures, we cannot modify.
3) What is the meaning of "Fig.12" before line 463.
Responses: we have removed the confused description.
4) the resolution of figures 23 and 27 needs to be improved. As you can see from the figures, some curves have wrong units.
Responses: we have removed the in-corrected description in these two figures, and they were re-drawn to get high resolution.
5) The low porosity and low permeability reservoir have been studied in the following literature, which may have some implications on this study.
Responses: We have add many necessary references to make this manusctipt much readability.
Reviewer 3 Report
I read this manuscript with interest and it is easy to see that the authors have done a lot of good work in this study and their research instruments and research strategies are worthy of replication, but from the perspective of the integrity and logic of the paper, the following suggestions are made.
1. In section 4.3, the better classification results are shown, if possible, can you further explain how the classification criteria in Table 1 were determined?
2. In section 5.2, the permeability calculation method is explored, and the permeability can be calculated based on the high correlation between the physical indicative parameters and R50. However, this section is too simple and the specific equation for permeability calculation is not given. It is suggested that this section should be improved.
3. It is suggested that the classification method proposed in the paper should be applied in the case study, and the accuracy of the classification should be verified by DST data, so as to make the study more complete.
Author Response
1) 1. In section 4.3, the better classification results are shown, if possible, can you further explain how the classification criteria in Table 1 were determined?
Responses: In Fig.11, we have carefully describe the establishment of classification criteria, these three typrs of core samples were classified based on porosity, permeability and pore structure (median pore throat radius, R50).
2) 2. In section 5.2, the permeability calculation method is explored, and the permeability can be calculated based on the high correlation between the physical indicative parameters and R50. However, this section is too simple and the specific equation for permeability calculation is not given. It is suggested that this section should be improved.
Responses: Thanks for your suggestive comments, Equ. 15 is added to imtroduce how to calculate permeability from porosity and R50.
3) It is suggested that the classification method proposed in the paper should be applied in the case study, and the accuracy of the classification should be verified by DST data, so as to make the study more complete.
Responses: Thanks for your positive comments, this classification is used to improve formation pore structure characterization and parameters prediction, and thus high-quality formation identification. They are helpfule in low permeability sandstone evaluation.
Reviewer 4 Report
Dear Authors,
I recommend that you highlight the effective methods used in your study to provide readers with a clear understanding of your approach. This will help readers to better comprehend the technical aspects of your research and the contribution it makes to the field. Additionally, it would be helpful to highlight the main results of your study in the abstract. This will give readers a sense of the significance of your research and the potential impact it may have.
Based on what you define the reservoir gas-bearing has low permeability when it ranges between 0.5 to 251 mD. I believe you can say low to average permeability.
In the geological setting, would you please add a stratigraphic column. Also please add the type of fault and develop a little bit of the tectonic event that affected the area of study.
In the reservoir petrophysical characteristics, you used thin sections, not slices. It will be preferable to add some images about your finding where you can highlight the type of mineral, type of pores, and contact between grains.
Which averages did you use to estimate porosity and permeability? Based on what you confirm that there is a complicated pore structure.
Based on your MICP data, please generated other figures related to pore size classification, J-Function, and Pore Throat Radii. I suggest adding in figures 6, 9, and 10 major and minor gridlines. Also, decrease the size of curves and points to help the reader to distinguish and read more easily the entry pressure and other information. In addition, I suggest converting Air/Mercury system to Brine/gas to be more realistic based on the component of the reservoir.
Please correct a typo in figure 9a (Type I instead of Tpye I).
For Figures 23, and 27, please improve the quality of the image. Also, add in the caption the description of each track.
Please increase the size of figures 25, 26
I suggest that you improve the quality of the conclusion by including the main results of your study. It is important to briefly summarize the main findings of your research in the conclusion to remind readers of the significance of your work. Furthermore, I recommend that you add a paragraph on future work or recommendations for further research. This section can help provide insights into the potential directions for future studies that can build upon your research.
Thank you!
Author Response
1) Based on what you define the reservoir gas-bearing has low permeability when it ranges between 0.5 to 251 mD. I believe you can say low to average permeability.
Responses: Permeability is indeed widely distributed from 0.5 to 251 mD du to the strong heterogeneity. low permeability is dominated, and the average permeability is only 7.86 mD, it belongs to low permeability.
2) In the geological setting, would you please add a stratigraphic column. Also please add the type of fault and develop a little bit of the tectonic event that affected the area of study.
Responses: We have modified Fig. 3 to add a stratigraphic column. In addition, the sedimentary environment of Pinghu Formation is added.
3)In the reservoir petrophysical characteristics, you used thin sections, not slices. It will be preferable to add some images about your finding where you can highlight the type of mineral, type of pores, and contact between grains.
Responses: Fig.s 4b and c are added to describe the facies type of our target formation.
4) Which averages did you use to estimate porosity and permeability? Based on what you confirm that there is a complicated pore structure.
Responses: Porosity is averaged by arithmetic mean method, and permeability is averaged by logarithmic mean. poor relationship between porosity and permeability displayed in Fig. 1, and wide permeability distribution exhibited the strong heterogeneity, and wide MICP distribution in Fig. 6 verified the complicated pore structure.
5) Based on your MICP data, please generated other figures related to pore size classification, J-Function, and Pore Throat Radii. I suggest adding in figures 6, 9, and 10 major and minor gridlines. Also, decrease the size of curves and points to help the reader to distinguish and read more easily the entry pressure and other information. In addition, I suggest converting Air/Mercury system to Brine/gas to be more realistic based on the component of the reservoir.
Responses: Thanks for your suggestive comments we have revised the manuscript following with your suggestions.
6) Please correct a typo in figure 9a (Type I instead of Tpye I).
Responses: I am sorry for the mistakes, we have removed it.
7) For Figures 23, and 27, please improve the quality of the image. Also, add in the caption the description of each track.
Responses: We have uploaded Fig.s 23 and 27. They were clear.
8) Please increase the size of figures 25, 26.
Responses: Thanks for your good comments, we have modified these two figures.
9) I suggest that you improve the quality of the conclusion by including the main results of your study. It is important to briefly summarize the main findings of your research in the conclusion to remind readers of the significance of your work. Furthermore, I recommend that you add a paragraph on future work or recommendations for further research. This section can help provide insights into the potential directions for future studies that can build upon your research.
Response: This manuscript is more of case studies. Several methods and techniques were used to improve formation evaluation. Hence, I think it was no need to give a future work.
Round 2
Reviewer 4 Report
Please improve the caption in the figure 4 and insert the scales in the thin sections.
Please insert minor gridline in all your graphs.
Please improve the figure 23, names are blurry and add comment for each track what it represents.
Do the same for figure 27
Author Response
1) Please improve the caption in the figure 4 and insert the scales in the thin sections.
Response: Done
2) Please insert minor gridline in all your graphs.
Response: Done
3) Please improve the figure 23, names are blurry and add comment for each track what it represents.
Response: We have revised figures 23 and 27 to make them much clearly, and we also upload the vector graph (tiff and pdf) that can be edited. Please find them in the attachment. we also have interpreted the physical significance for every logging curve through line 421 to 434. Please find them, thank you!
4) Do the same for figure 27
Response: Done